# FRODO: Free rejection of out-of-distribution samples: application to chest x-ray analysis

**Erdi Çallı**[1]                                                          ERDI.CALLI@RADBOUDUMC.NL
**Keelin Murphy**[1]                                                  KEELIN.MURPHY@RADBOUDUMC.NL
**Ecem Sogancioglu**[1]                                            ECEM.LAGO@RADBOUDUMC.NL
**Bram van Ginneken**[1]                              BRAM.VANGINNEKEN@RADBOUDUMC.NL

[1] *Diagnostic Image Analysis Group, Radboud University Medical Center*

## Abstract

In this work, we propose a method to reject out-of-distribution samples which can be adapted to any network architecture and requires no additional training data. Publicly available chest x-ray data (38,353 images) is used to train a standard ResNet-50 model to detect emphysema. Feature activations of intermediate layers are used as descriptors defining the training data distribution. A novel metric, FRODO, is measured by using the Mahalanobis distance of a new test sample to the training data distribution. The method is tested using a held-out test dataset of 21,176 chest x-rays (in-distribution) and a set of 14,821 out-of-distribution x-ray images of incorrect orientation or anatomy. In classifying test samples as in or out-of distribution, our method achieves an AUC score of 0.99.

## 1. Introduction

We use deep learning in medical imaging to identify anatomy or pathology on specified image types. However, neural networks are not fail-safe if presented with images from different anatomical regions or largely excluding the region of interest. We refer to such data as out-of-distribution (OOD) samples and propose a metric to reject them using a standard neural network trained only on 'in-distribution' images of the anatomical region of interest. We define FRODO using the Mahalanobis distance (Mahalanobis, 1936) of intermediate feature activations as a metric that describes the distance of a given sample from the training data distribution. We use a model trained with frontal chest x-rays (CXR) to demonstrate that this metric can be used to detect non-CXR and lateral CXR (i.e. OOD samples) successfully. We compare this method to a baseline OOD detection method (Hendrycks and Gimpel, 2017). We show that our method outperforms the baseline by a significant margin.

## 2. Methods

The feature activations of intermediate layers in a deep learning model are known to be excellent descriptors of the training data (Donahue et al., 2013). In this work we hypothesize that feature activations from a homogeneous set of training data (CXRs) have a specific distribution and that OOD samples will produce feature activations some distance away. We use the Mahalanobis distance (Mahalanobis, 1936) to measure the distance between a

distribution of previous observations and a new sample. This method has also been used in (Denouden et al., 2018) and (Lee et al., 2018). (Denouden et al., 2018) investigates its use in the context of an auto-encoder. (Lee et al., 2018) shows that this metric is useful in a class-conditional setting. Both methods work with tasks involving natural images. Our method differs by restricting the in-distribution images to an anatomical region of interest and being independent of the task (e.g. classification of a pathological condition) of the model.

We train a (pre-trained) ResNet-50 (He et al., 2016) model on the emphysema labels of the ChestXray14 dataset (Wang et al., 2017). Our dataset consists of 38,353 training (854 positive), 4,625 validation (88 positive) and 21,176 test (436 positive) samples. This dataset consists of frontal CXRs with unique patients in each split. The parameters that were used to train this model are described in our previous work (Çallı et al., 2019).

From the trained network, we obtain feature activation distributions from specific layers. The distribution parameters are used to calculate the Mahalanobis distance of a test sample, on each of these layers. Because of the residual architecture of ResNet we consider only the deepest layer of residual blocks that have the same output dimensions. Table 1 describes the output dimensions of the layers **L1**-**L5** that are used.

Table 1: Output dimensions of the layers from ResNet-50 used for feature activation analysis. **L1** is the first convolutional layer. **L[2-5]** refer to the deepest residual layers that have the specified dimensions.

| **L1:** 112x112x64 | **L2:** 56x56x256 | **L3:** 28x28x512 | **L4:** 14x14x1024 | **L5:** 7x7x2048 |
| --- | --- | --- | --- | --- |

## 3. Experiments

We use an OOD dataset of x-ray images collected from our institute. This dataset consists of 14,821 images including lateral CXRs, x-rays of other body parts (e.g. head, arm, leg, abdomen), and frontal CXRs which are very noisy or exclude a significant proportion of the lung volume. The test dataset of 21,176 images from ChestXray14 is used as the in-distribution testing data. Using thresholds on the FRODO distances we classify all test samples as in- or out-of-distribution and plot ROC curves of the results. We compare our method with the baseline OOD detection method (Hendrycks and Gimpel, 2017). This work assumes that the in-distribution samples have higher prediction confidence than the OOD samples. Therefore, they use the probability of the most likely class for the classification of in-distribution samples. We report the AUC of each method.

## 4. Results

For the emphysema classification task our model achieves 0.854 AUC on the test dataset. In Figure 1, we show ROC curves for in and out-of-distribution classification using FRODO and the baseline method (Hendrycks and Gimpel, 2017). FRODO outperforms the baseline by a significant margin. Feature activations on layer **L3** were most effective at distinguishing

OOD samples with an AUC of 0.99, compared to 0.80 for the baseline method. In Figure 1 we show sample results at 99% sensitivity to OOD detection.

Figure 1: Left: ROCs for different methods. Right: Some examples of incorrectly accepted, incorrectly rejected, and correctly rejected samples in first, second, and third rows, respectively.

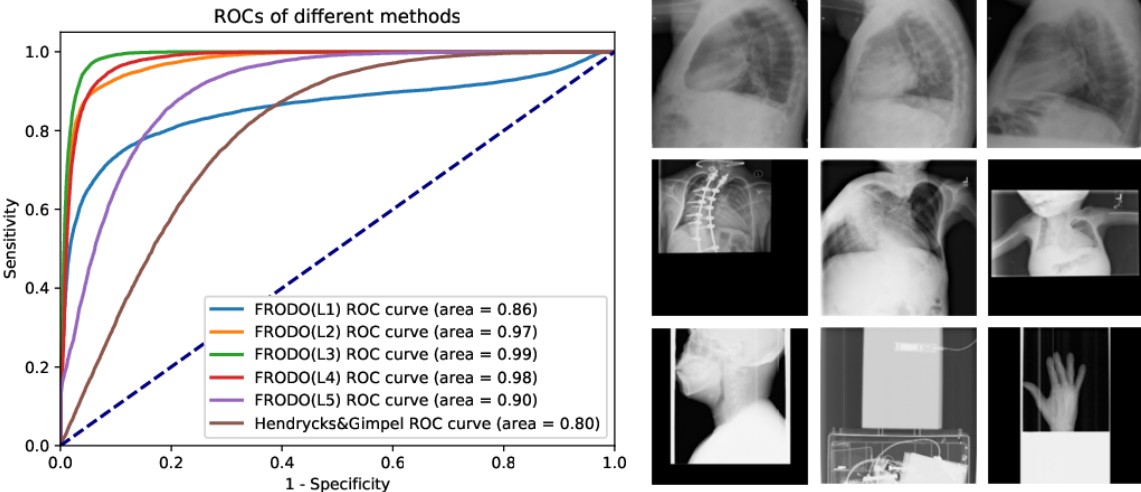

## 5. Discussion

Rejection of OOD samples is a key step towards moving medical image analysis into the clinic, ensuring that algorithms do not provide meaningless scores and instead reject unsuitable data for manual review. In this study, we have demonstrated FRODO, a method to classify OOD samples obtaining 0.99 AUC. This method is free, it neither requires any change on the neural network structure, nor any OOD training examples. One limitation of this study is the differing sources of the in and out-of-distribution samples, which will be addressed in future work.

The FRODO method has most difficulty distinguishing lateral CXRs as OOD although it still classifies 90% of these correctly. At 99% sensitivity for detection of OOD samples no other type of OOD image was incorrectly accepted. Of those CXR images incorrectly rejected, we observe that they include many examples of poor collimation, artefact or the presence of foreign objects.

(Shwartz-Ziv and Tishby, 2017) suggests that a trained neural network converges to an encoder-decoder pair, where the encoder is optimized for a good feature representation of the input and the decoder is optimized to perform the task using these features. This suggests that the deepest encoder layer has the most informative set of features describing an input. Our results show an increase in performance towards **L3** and a decrease in performance afterwards. This suggests that **L3** is the deepest encoding layer and has the features that are the most representative of the input.

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
