# OpenReview forum: "FRODO: Free rejection of out-of-distribution samples: application to chest x-ray analysis"
_MIDL.io/2019/Conference/Abstract — MIDL Abstract 2019_

### Official Review · AnonReviewer2 · 2019-04-24
**Overall interesting work and I am looking forward to discuss this at MIDL.**

**Rating:** 4
**Confidence:** 3

**Review:**

This abstract discusses a simple method to build an out-of-distribution detection approach by exploiting the Mahalanobis distance to measure the distance between a distribution of previous observations and a new sample for a trained network.

"This work assumes that the in-distribution samples have higher prediction confidence" -- how far does this assumption go?  Would it be possible to build a simple linear classier or histogram-based to detect these samples in this dataset with similar performance? It would be interesting to evaluate this on a healthy/disease set of similar anatomical presentation instead of wildly random out-of-distribution samples as shown in this work. Is an adversarial attack possible here? I would assume that random noise patterns will also robustly be identified as OOD but how's about adversarial attack patterns? How much would it take to make an OOD sample and in-distribution sample.

Overall interesting work and I am looking forward to discuss this at MIDL.

---

### Official Review · AnonReviewer1 · 2019-04-25
**simple but useful**

**Rating:** 3
**Confidence:** 3

**Review:**

This paper proposes a method to detect outliers in a dataset, meaning images of other body parts, etc. This is particularly interesting in big data studies extracting data from hospital PACS, where DICOM headers cannot always be trusted. The method simply builds a Gaussian distribution of activations of units in a certain layer when the input scans are not outliers, and then evaluates a new scan under that distribution to decide whether it is an outlier or not. The results seem convincing. My only suggestion is that the authors may want to train a more complex classifier from the activations, as it seems very unlikely that their distribution is Gaussian.

---

### Decision · Program_Chairs · 2019-05-06
**Acceptance Decision**

Accept